# Adlay Consumption Combined with Suspension Training Improves Blood Lipids and Pulse Wave Velocity in Middle-Aged Women

**DOI:** 10.3390/healthcare11101426

**Published:** 2023-05-15

**Authors:** Chae Kwan Lee, Jae-Hoon Lee, Shuho Kang, Min-Seong Ha

**Affiliations:** 1Department of Physical Therapy, Catholic University of Pusan, 57 Oryundae-ro, Geumjeong-gu, Busan 46252, Republic of Korea; 2Department of Sports Science, College of the Arts and Sports, University of Seoul, 163 Seoulsiripdaero, Dongdaemun-gu, Seoul 02504, Republic of Korea; 3Graduate School, Busan University of Foreign Studies, 65 Geumsaem-ro 485-gil, Geumjeong-gu, Busan 46234, Republic of Korea

**Keywords:** adlay, grains, suspension training, pre-menopausal women, blood lipids, physical fitness, supplement

## Abstract

Middle-aged women have an increased risk of chronic degenerative diseases and reduced physical strength, which can lead to decreased vascular function and an increased risk of cardiovascular disease. However, these problems can be treated or prevented with healthy nutrition and regular exercise. We focused on these benefits as recent studies have reported the potential synergistic effects of suspension training and nutrition. Therefore, in this study, we investigated the effects of 12 weeks of adlay intake and suspension training on improvements in body composition, physical fitness, blood lipids, and arterial stiffness in middle-aged women. Neither the adlay + suspension exercise (ASEG) nor suspension exercise groups (SEG) showed significant changes in body composition. Nonetheless, with respect to physical fitness, there was a difference in time among all variables except flexibility, though the ASEG showed a more significant effect than the SEG. Regarding blood lipids, significant interaction effects were found for triglycerides, high-density lipoprotein cholesterol, and low-density lipoprotein cholesterol, while only the ASEG significantly improved these parameters. Furthermore, pulse wave velocity was only significantly decreased in the ASEG. In summary, performing suspension exercises for 12 weeks improved physical strength in middle-aged women. Additionally, when adlay was consumed simultaneously, blood lipids and arterial stiffness were improved.

## 1. Introduction

Women are at a higher risk of disease than men owing to their physiological, social, and environmental characteristics. Therefore, although the life expectancy of women is longer than that of men, the healthy lifespans of women tend to be shorter [1,2]. In addition, their muscle mass decreases with age, resulting in muscle weakness, particularly in menopausal women. Thus, middle-aged women are at a high risk of chronic degenerative diseases [3] and a decline in physical fitness. Additionally, aging causes a decline in vascular function in addition to an increase in the risk of cardiovascular disease due to changes in blood pressure and lipids and risk factors of obesity and hyperlipidemia [4,5]. Although exercise and a healthy diet are essential in mitigating these risks, more efficient methods are necessary to reduce the incidence of aging-related diseases, especially in women.

Adlay (Coix lachryma-jobi L. var. ma-yuen) is a cereal that is enriched in crude protein, crude fat, vitamins B1 and B2, iron, and calcium, making it a popular dietary supplement [6]. Adlay has also been verified as a healthy food and has a long history of use in traditional medicine. It was previously shown to regulate the expression of tumor necrosis factor-alpha, which is involved in adipocyte metabolism, thereby reducing hyperlipidemia through the regulation of body weight and adipose tissue [7]. Furthermore, adlay has been found to prevent vascular diseases such as sclerosis and coronary artery disease [8]. Considering that the physiological activity and nutritional effects of these grains have been verified, many nutritional researchers have set the goal of “maintaining a healthy life through natural foods” [9].

In addition to a healthy diet, regular exercise has been proven to be capable of both treating and preventing chronic diseases in middle-aged women. Nevertheless, this combination is rarely practiced in everyday life to obtain these health benefits. Therefore, it is crucial to investigate more efficient exercise methods. Suspension training is a type of total-body resistance exercise that controls the resistance arising from one’s own body weight via a tool connected to a two-pronged string from one axis. Suspension training has emerged as an effective exercise method for increasing muscle activity to a greater extent than in general resistance exercise [10] while also improving body weight and blood lipids [11]. Additionally, by using suspension training, resistance can be applied to the whole body rather than individual body parts, resulting in the use of more muscles during exercise owing to the high motor unit mobilization rate. Therefore, suspension training is widely considered a very efficient exercise method [12,13].

One recent study, which reported the synergistic effect of suspension training in combination with nutritional supplementation [14], highlighted the need to combine suspension training with a nutritional approach. Therefore, we hypothesized that combining the intake of adlay and suspension training would have a synergistic effect on the health and fitness of middle-aged women when compared to the use of suspension training alone. Therefore, we conducted this study to investigate the changes in body composition, physical fitness, blood lipids, and arterial stiffness due to intervening adlay supplements and suspension training for 12 weeks in middle-aged women.

## 2. Materials and Methods

### 2.1. Participants

Using G-power, version 3.1 for Windows (Kiel University, Kiel, Germany), a power analysis with a desired effect size of 0.25 (default), significance level of 0.05, and power of 0.50 determined that a sample size of 21 was required for this study [15]. Using community flyers, we recruited middle-aged, pre-menopausal women between the ages of 40 and 50 living in Busan, South Korea. After eliminating potential participants according to the following exclusion criteria: (1) uncontrolled high blood pressure and diabetes, (2) allergic reaction to the consumption of adlay, (3) suffering from cardiovascular or musculoskeletal disease, (4) treated with psychiatric drugs or herbal medicines within the last two months, and (5) any individual considered to be unsuitable for the study based on an assessment by the research director, a total of 24 women were selected for the study.

This study was conducted with the approval of the Bioethics Committee of the Catholic University of Korea and in line with the Declaration of Helsinki (CUPIRB/2022-056). The purpose of this study was fully explained to all study participants, and consent was obtained before the experiment began. All study participants completed the experiment without dropping out. The participants’ physical characteristics at baseline are summarized in Table 1.

### 2.2. Study Design

Using a randomized, double-blind clinical trial, we examined the effects of adlay intake and/or suspension exercise for 12 weeks. The participants were randomly and equally divided among the adlay + suspension exercise group (ASEG; *n* = 8), suspension exercise group (SEG; *n* = 8), and control group (CON; *n* = 8). All measurements were taken at two time points: before and after the experiment. The study design is shown in Figure 1.

### 2.3. Suspension Training Program

The suspension exercise program consisted of 10 min of warm-up and cool-down exercises in addition to 40 min of the main exercise three times a week for 12 weeks. Exercise intensity was determined using the rating of perceived exertion (RPE) index, which was regulated with a wearable heart rate monitor (Polar RS400sd; model APAC, Polar, Bethpage, NY, USA) [16]. The program was set to RPE 11–12 (40–50% heart rate reserve [HRR]) for weeks 1–2, RPE 13–14 (50–60% HRR) for weeks 3–8, and RPE 15–16 (60–70% HRR) for weeks 9–12. The exercise intensity was set in accordance with Dawes [17]. The details of the suspension exercise program are shown in Table 2.

### 2.4. Adlay Intake and Composition

Based on the results of a previous study, which used rats that were administered 50 mg of adlay per 100 g of body weight [18], we set the daily intake of adlay at 30 g, considering the average weight of the study participants to be approximately 60 kg. Participants ingested 15 g of adlay twice: once before and once after exercising. The exercise was performed three times per week. Therefore, the participants consumed the adlay a total of six times per week, which is equivalent to 180 g per week. Adlay powder (J Foods; 4.08 ± 0.03% water, 72.68 ± 0.02% carbohydrate, 5.16 ± 0.04% crude fat, 15.65 ± 0.08% crude protein, 1.55 ± 0.02% crude ash, Pukyong Feed and Foods Nutrition Research Center, 2015) was provided to the participants using a quantified spoon. Additionally, the study participants did not consume any other dietary supplements or drugs during the study period.

### 2.5. Blood Sampling

All study participants were asked to fast for at least 8 h the day before blood collection. Between 8 a.m. and 10 a.m., a clinical pathologist collected 15 mL of blood from the participants’ forearms vein using a spasm blood collection tube and needle. The collected blood was then centrifuged at 3000 rpm for 10 min, using a Combi-514R (Hanil, Seoul, Republic of Korea), with the serum being used for analysis.

Total cholesterol (TC) and triglycerides (TGs) were analyzed via enzyme colorimetry using automatic analyzers and the contained reagents from Siemens (ADVIA-1650; Norcross, GA, USA). High-density lipoprotein (HDL) and low-density lipoprotein (LDL) cholesterol were also analyzed via the elimination/catalase method, using the ADVIA-1650 system (Siemens, Berlin, Germany) in accordance with the manufacturer instructions.

### 2.6. Body Composition and Fitness Test

After removing all precious metals, the body composition of each study participant was measured while the participant wore simple clothes. A physical fitness test was also performed. Body composition was calculated based on the participant’s height, weight, and lean mass, which were automatically measured using the X-SCAN PLUS II system (JAWON Medical, Seoul, Republic of Korea). The following four items were measured for the physical fitness assessment: muscular strength (grip strength), muscular endurance (sit-ups), flexibility (sit and stretch; sitting and forward bending), and cardiorespiratory endurance (20 m round-trip endurance run).

Grip strength was measured while the participant was holding a grip force meter (VA-HD8008, Seoul, Republic of Korea) with four fingers (excluding the thumb) by adjusting the handle width, naturally lowering the arm, and ensuring that the grip force meter did not come into contact with the body. After measuring the grip force twice, the highest value was recorded in units of 0.1 kg. Additionally, during sit-ups, the participant was required to use their abdominal strength to raise their upper body while crossing their arms over their chest for 60 s. For sitting and bending forward, the participant sat down and slowly bent their upper body before extending their hand under the scale of the measuring instrument (DW-704, Goyang, Republic of Korea), and then recording the scale at the point at which the fingertips reached for approximately 2 s. The forward bend was performed twice, with the best of two attempts recorded in units of 0.1 cm. In the 20 m round-trip endurance run, the running was repeated by covering a distance of 20 m in response to a beep with a set time interval. Both feet had to completely pass the 20 m line before the beep, and the maximum number of repeated runs possible was recorded.

### 2.7. Arterial Stiffness

The carotid–femoral pulse wave velocity (cf-PWV) is considered the “gold-standard” measurement of arterial stiffness. The cf-PWV is considered representative of the PWV for the entire aorta. The PWV was measured as an index of arterial stiffness in the carotid–femoral arteries using a SphygmoCor (AtCor Medical, Denistone, NSW, Australia) while the participant was in the supine position. The SphygmoCor is a tonometry-type pulse-wave measuring device that can detect pressure pulse waves in a non-invasive manner. All measurements were made in accordance with the Clinical Application of Arterial stiffness guidelines, Task Force III [19].

The pulse-wave propagation velocity was automatically recorded using the automatic software of the measuring device. The measuring device recorded the pulse waves from both sides of the artery when the pulse wave progressed from the carotid artery to the brachial artery; the interval was further measured using a tape measure. The automatic software of the measuring instrument then divided the distance (L) between the two measures according to the time difference (Δt) between the pulse waves recorded at both points. 

### 2.8. Statistical Analysis

All data were analyzed using IBM SPSS 27.0, and the mean and standard deviation were calculated from the measured item values. To investigate the effects of the combined intervention of adlay and suspension exercise for 12 weeks on blood lipids, physical strength, and arterial stiffness, a 3 × 2 two-way repeated-measures analysis of variance was performed with treatment (ASEG, SEG, CON) and time (pre, post) as independent variables. Furthermore, Duncan’s test was used for a post-hoc analysis in pairwise comparisons between groups at the end of the experiment. The statistical significance level was set at *p* < 0.05.

## 3. Results

### 3.1. Effect of Adlay and Suspension Training for 12 Weeks on Body Composition

Figure 2 and Appendix A both show the changes in body composition among the three groups. Body mass index (BMI) was significantly increased from 22.1 ± 2.3 kg/m^2^ to 22.5 ± 3.0 kg/m^2^ in the CON group (*p* < 0.05), while no significant effect was observed for time, group, or the time × group interaction. Additionally, there was no significant difference in lean body mass (LBM) for any group; no significant effects were observed from time, group, or the time × group interaction.

### 3.2. Effect of Adlay and Suspension Training for 12 Weeks on Physical Fitness

Figure 3 and Appendix A show the changes in physical fitness variables for the three groups after the 12-week intervention. Grip strength was increased from 27.3 ± 2.9 kg to 29.4 ± 2.8 kg (*p* < 0.01) in the ASEG, compared to increasing from 27.4 ± 3.0 to 28.0 ± 3.0 (*p* < 0.05) in the SEG. A significant effect was found for both time and the time × group interaction. Furthermore, a post-hoc analysis showed that grip strength was increased in the ASEG compared with the SEG and CON group. The number of sit-ups performed in 60 s increased from 24.1 ± 6.5 to 29.1 ± 7.8 (*p* < 0.01) in the ASEG and from 24.5 ± 5.7 to 27.4 ± 5.8 (*p* < 0.01) in the SEG. Significant effects were found for time, group, and the time × group interactions. Additionally, post-hoc analysis showed that the sit-up ability of the ASEG and SEG were improved compared with that of the CON group after the 12-week intervention. There was no significant difference in the sit and reach test (forward bend) over time for any group, and there was no significant effect of time, group, or time × group interaction. The number of laps performed in the 20 m multi-stage endurance run was increased to 21.8 ± 4.1 from 19.1 ± 4.1 (*p* < 0.01) in the ASEG and to 21.4 ± 6.4 from 19.8 ± 6.5 (*p* < 0.05) in the SEG. Significant effects were found for time and the time × group interactions. In addition, a post-hoc analysis showed that the participants in the ASEG and SEG could run more laps than those in the CON group after the 12-week intervention.

### 3.3. Effect of Adlay and Suspension Training for 12 Weeks on Blood Lipid Levels

Figure 4 and Appendix A both show the changes in blood lipids among the three groups after the 12-week intervention. There was no significant difference in TC levels for any group after the intervention, and there were no significant effects from time, group, or time × group interaction. Additionally, TG levels significantly decreased from 80.6 ± 8.8 mg/dL to 77.8 ± 9.7 mg/dL in only the ASEG (*p* < 0.01), with time having a significant effect. HDL cholesterol was significantly increased from 50.3 ± 6.0 mg/dL to 53.6 ± 5.8 mg/dL in only the ASEG (*p* < 0.05), and there was again a significant effect of time. Similarly, LDL cholesterol showed a significant decrease in the ASEG, from 115.6 ± 6.8 mg/dL to 110.6 ± 7.8 mg/dL (*p* < 0.01), as well as in the SEG, from 115.3 ± 8.3 mg/dL to 112.5 ± 7.8 mg/dL (*p* < 0.05). There were significant effects from time and time × group here. Post-hoc analysis subsequently showed that the LDL cholesterol level in the CON group was higher than that in the ASEG.

### 3.4. Effect of Adlay and Suspension Training for 12 Weeks on Arterial Stiffness

Figure 5 and Appendix A show the changes in arterial stiffness after 12 weeks among the three groups. The PWV of the ASEG was decreased from 8.3 ± 0.7 to 7.3 ± 0.7 (*p* < 0.01), with significant effects from both time and time × group. A post-hoc analysis also showed that the PWV values of the SEG and CON group were higher than those of the ASEG after the experiment.

## 4. Discussion

This study was conducted based on the hypothesis that synergistic effects on improved fitness and blood lipid parameters can be achieved if suspension training is performed alongside the consumption of adlay by middle-aged, pre-menopausal women. Our results subsequently confirmed that 12 weeks of suspension training improved physical fitness variables (grip strength; sit-ups, sit and reach; 20 m multi-stage running). In addition, synergistic effects of suspension training in combination with adlay consumption were found to have improved blood lipids (TG, HDL-C, and LDL-C) and PWV.

We found contradictory results for the effect of adlay ingestion alongside suspension training on BMI and LBM compared with those of a previous study [20]. In another previous study, waist circumference and blood pressure were observed to have decreased in participants who performed suspension training for eight weeks [21]. Additionally, suspension training was found to have been an effective exercise method for improving body composition in overweight women, as this exercise resulted in a decrease in the percentage and amount of body fat [22]. The conflicting results in the present study may have been due to several factors. First, as this study was conducted using only pre-menopausal women, the conditions of the exercise program differed from those of previous studies that found an improvement in body composition. Second, considering that men can perform higher-intensity exercises better than women, the participants’ body composition would have changed more when both men and women participated in the study. Third, these differences among studies may have been due to the specific nutritional effects of consuming adlay. Nonetheless, if the intensity of the exercise program had been increased to match that of previous studies, participants with weak physical strength and beginners may have experienced pain due to joint damage. Therefore, the conditions of the exercise program should be carefully determined [23].

One previous study using suspension training in healthy adults three times a week for eight weeks found a reduction in disease risk by improving not only muscle strength and endurance but also predictors of cardiovascular disease, such as waist circumference and blood pressure [21]. Another previous study that focused on underweight women found similar results, indicating that a high body weight is not a requirement for suspension training to be effective [24]. However, there was no significant difference in flexibility, which could be explained by the fact that suspension exercise focuses more on strength and muscular endurance than flexibility.

In this study, grip strength and the 20 m multi-stage running test were both more greatly effect in the ASEG than the SEG. It was assumed that increasing glycogen during training results in increased muscle mass, thus enhancing strength and endurance [25]. Furthermore, when grains are consumed after exercise, the rate of glycogen re-accumulation in the muscle and liver increases, allowing for an effective recovery [26]. Therefore, these results are believed to have been achieved by consuming adlay alongside suspension training.

There are six functional ingredients in adlay, including oils, polysaccharides, phenols, coixol, phytosterols, and resistant starch, which have more than 10 documented health benefits. Therefore, adlay has been widely used as a health food and medicine [27], with the antioxidant activity of adlay being well-known. Adlay powder was also reported to have improved blood lipids by improving TC, TG, HDL-C, and LDL-C levels, with this effect being more significant among patients with dyslipidemia, especially middle-aged patients [28]. Thus, adlay consumption could largely account for the improvement in blood lipids observed in the ASEG in this study.

Atherosclerosis is closely related to coronary heart disease, and high cholesterol is known to contribute to arteriosclerosis [29]. The administration of fermented adlay to hamsters for four weeks was found to have been able to regulate the oxidative stress caused by hyperlipidemia while also improving cholesterol levels [30]. Similarly, feeding adlay to obese rats for 12 weeks lowered their systolic blood pressure [31].

One study in young adults showed that suspension training for eight weeks resulted in improved blood pressure and cardiorespiratory fitness [32]. Collectively, these studies suggest that suspension training could improve blood pressure, while adlay could improve blood lipids and blood pressure in both animal models and humans. Furthermore, the improvement of PWV, TG, and HDL-C in the ASEG observed in this study may have been related to the mechanism described above.

One potential limitation of this study is the small number of study subjects. Considering that the effect size was 50%, it is difficult to generalize the results of this study. However, these results may provide evidence for complex treatment to improve the fitness and health of middle-aged women. In addition, considering that we focused only on pre-menopausal middle-aged women, it is difficult to generalize the results of this study for men or women of other age groups.

## 5. Conclusions

In summary, we found no significant changes in body composition for either the ASEG or SEG. However, this intervention had a positive effect on all physical fitness parameters other than flexibility. Additionally, this effect was greater in the ASEG than the SEG. Among the measured blood lipid variables, TG, HDL-C, and LDL-C all showed a significant interaction effect, while only the ASEG showed significant improvement in these parameters when comparing results from before and after the experiment. Lastly, PWV, as an arterial stiffness index, showed that there were substantial effects from time and the time × group interaction. When comparing the analysis results before and after the experiment, we found a significant reduction of PWV in only the ASEG. Therefore, performing suspension exercises for 12 weeks improved the physical strength of middle-aged women. Additionally, when adlay was consumed simultaneously, blood lipids and arterial stiffness improved.

## Figures and Tables

**Figure 1 healthcare-11-01426-f001:**
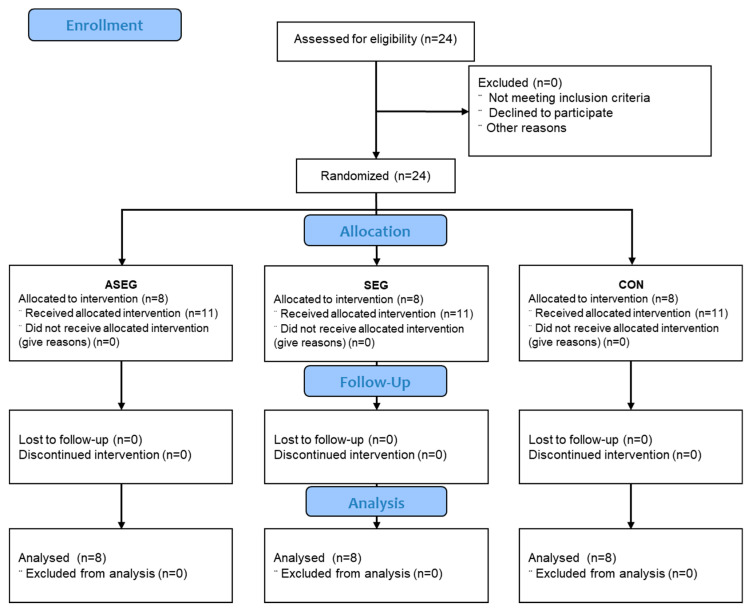
CONSORT flow diagram. ASEG: adlay + suspension exercise group; SEG: suspension exercise group; CON: control.

**Figure 2 healthcare-11-01426-f002:**
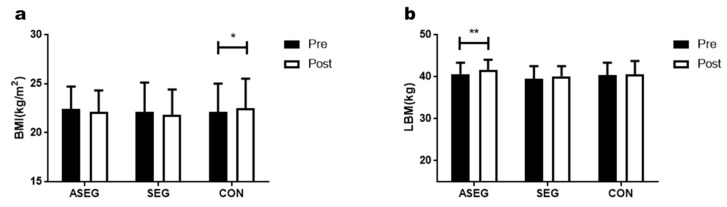
Effect of 12 weeks of suspension training and adlay intake on body composition in middle-aged women. After the intervention, compared with the baseline values, (**a**) BMI levels were increased in the CON group while (**b**) LBM levels were decreased in the ASEG. Data are presented as the mean ± standard deviation; * *p* < 0.05, ** *p* < 0.01 before vs. after intervention.

**Figure 3 healthcare-11-01426-f003:**
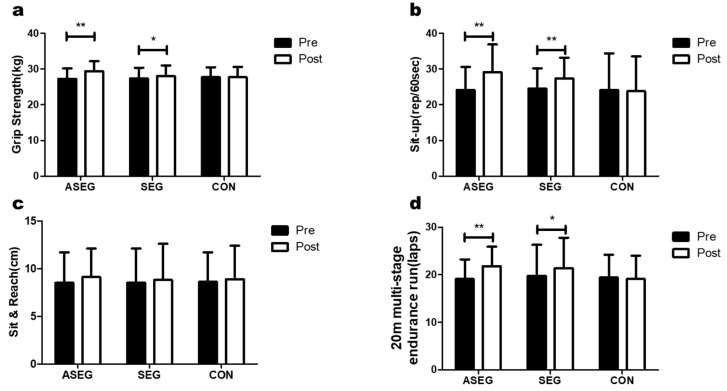
Effect of 12 weeks of suspension and adlay intake on fitness test parameters in middle-aged women. After the intervention, compared with the baseline values, (**a**) grip strength was increased in the ASEG and SEG, (**b**) the number of sit-ups increased in the ASEG and SEG, (**c**) there were no significant differences in all groups for the sit and reach test, and (**d**) the number of laps performed in the 20 m multi-stage endurance run was increased in the ASEG and SEG. Data are presented as the mean ± standard deviation; * *p* < 0.05, ** *p* < 0.01 before vs. after the intervention.

**Figure 4 healthcare-11-01426-f004:**
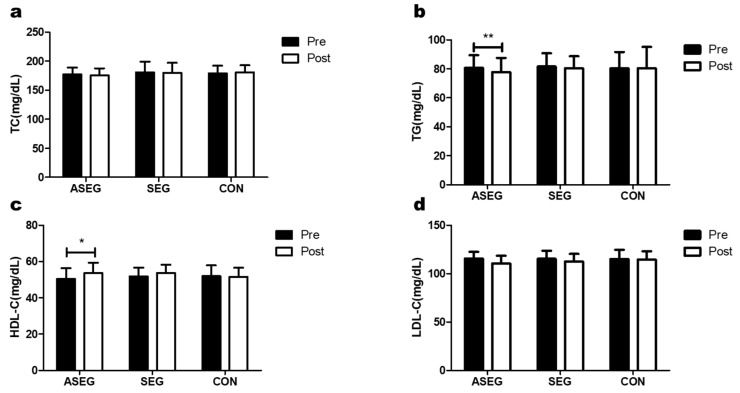
Effect of 12 weeks of suspension and adlay intake on blood lipids in middle-aged women. After the intervention, compared with the baseline values, (**a**) total cholesterol (TC) and (**d**) low-density lipoprotein cholesterol (LDL-C) showed no significant differences among all groups. (**b**) Triglyceride (TG) levels were decreased in the ASEG, whereas (**c**) high-density lipoprotein cholesterol (HDL-C) levels were increased in the ASEG. Data are presented as the mean ± standard deviation; * *p* < 0.05, ** *p* < 0.01 before vs. after intervention.

**Figure 5 healthcare-11-01426-f005:**
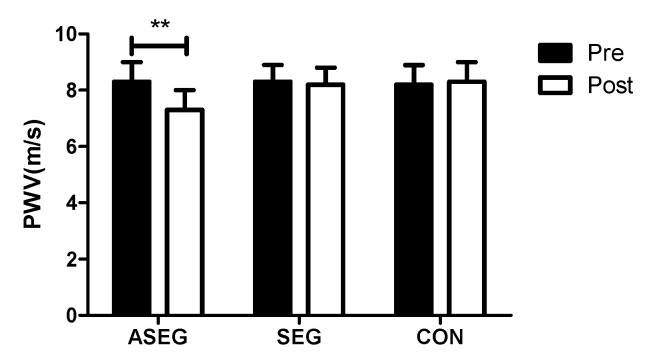
Effect of 12 weeks of suspension and adlay intake on arterial stiffness in middle-aged women. After the intervention, pulse wave velocity was decreased in the ASEG compared with the baseline values. Data are presented as the mean ± standard deviation; ** *p* < 0.01 before vs. after intervention.

**Table 1 healthcare-11-01426-t001:** Baseline characteristics of the participants.

Variables	ASEG (*n* = 8)	SEG (*n* = 8)	CON (*n* = 8)
Age (years)	47.4 ± 1.6	47.1 ± 1.5	47.5 ± 1.4
Height (cm)	158.3 ± 3.8	159.5 ± 3.9	158.3 ± 2.9
Weight (kg)	62.6 ± 3.2	61.1 ±3.7	62.4 ± 3.6
BMI (kg/m^2^)	22.4 ± 2.3	22.1 ± 3.0	22.1 ± 2.9

Values are presented as mean ± standard deviation. ASEG: adlay + suspension exercise group; SEG: suspension exercise group; CON: control group; BMI: body mass index.

**Table 2 healthcare-11-01426-t002:** Suspension exercise program.

Week	Order	Exercise	Intensity		Frequency
	Warm-up (10 min)	Static stretching SMR			
1–2	Main exercise (40 min)	Suspension plank Suspension push-up Suspension balance lunge Suspension pike Suspension row Suspension jump squat Suspension arm curl Suspension Y deltoid fly	3 sets	40–50% HRR (RPE 11–12)	3 times/ week
3–8			4 sets	50–60% HRR (RPE 13–14)	
9–12			5 sets	60–70% HRR (RPE 15–16)	
	Cool-down (10 min)	Static stretching SMR			

SMR: self-myofascial release; RPE: rating of perceived exertion; HRR: heart rate reserve.

## Data Availability

The authors declare that all data and materials are available to be shared on a formal request.

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
