# Peer review of "Adlay Consumption Combined with Suspension Training Improves Blood Lipids and Pulse Wave Velocity in Middle-Aged Women"

_healthcare, 2023, doi:10.3390/healthcare11101426_

Round 1

Reviewer 1 Report

The authors hypothesized that intake of adlay and suspension training in middle-aged women would be more effective for body composition, physical strength, blood lipids, and arterial stiffness than suspension training alone, and tested the hypothesis in a randomized, double-blind clinical trial. The research paper is very interesting, and it can be judged that the structure of the paper is complete. I suspect the authors will need some minor corrections.

1. It can be modified more clearly and concisely in the purpose of research, which is the last part of the introduction.

2. Please check the grammatical part again from the spell check in the body as a whole.

3. There are some awkward parts of English expressions. Therefore, you need to get checked again by native speakers.

It's a simple correction, but your research paper will be more complete when you supplement the relevant part.

Author Response

Response to Reviewers

Manuscript Title: Adlay consumption combined with suspension training improves blood lipids and pulse wave velocity

Editorial Board Member,

Healthcare

We appreciate the positive comments from the reviewers. Based on your comments, our manuscript was finally revised. We eagerly look forward to publishing the revised manuscript in the Healthcare.

To Reviewers:

We thank the reviewer for important suggestions on our manuscript. We carefully read each reviewer's comments and revised the manuscript accordingly. Each comment is outlined below, point by point. Therefore, we believe that the revised version of the manuscript is a significant improvement over the initial submission.

Reviewer 2 Report

This original research paper by Lee et al. investigates the effects of Adlay consumption combined with suspension training on the body composition, physical fitness, blood lipids, and arterial stiffness in middle-aged women. The reviewer thinks the article could be improved before publication and proposes comments below to help the authors.

Major comments: 

The reviewer suggests adding "middle-aged women" in the article's title since the study only investigates the effects of their treatment in middle-aged women. 

All terms should be explained in the abstract. ASEG and SEG when describing the study in lanes 16-18, and TG, HDL-C, LDL-c, and PWV in lane 22. 

The reviewer suggests showing all individual data points instead of bar graphs and all post-hoc analyses in tables. This is good practice and could help the reader better appreciate the data range differences noticed/claimed by the authors in Figures 2a, 3a, 3b, 3d, 4b, 4c, and 5.

See article 14 in the references: Samadpour Masouleh, S.; Bagheri, R.; Ashtary-Larky, D.; Cheraghloo, N.; Wong, A.; Yousefi Bilesvar, O.; Suzuki, K.; 370 Siahkouhian, M. The effects of TRX suspension training combined with taurine supplementation on body composition, 371 glycemic and lipid markers in women with type 2 diabetes. Nutrients 2021, 13, 3958.

 Minor comments:

There is probably a typo on lane 14. The authors probably meant "These common chronic diseases..." instead of "These chronic diseases common..."

The BMI graph could start around 10-1. No one could have a BMI between 0 and 15, and individuals with a BMI below 18 are considered underweight. The current presentation (bar graph starting at 0) does not allow the reader to appreciate the data range.

Author Response

(The authors gave the same response as above.)

Reviewer 3 Report

The authors hypothesized that combining the intake of adlay and suspension training would have a synergistic effect on the health and fitness of middle-aged women compared to the use of suspension training alone. To test this hypothesis, authors evaluated improvements in body composition, physical fitness, blood lipids, and arterial stiffness by intervening adlay supplements and suspension training for 12 weeks in middle-aged women. The purpose of present study was to assess the combined Adlay supplementation and suspension training on  body composition, physical fitness, blood lipids, and arterial stiffness in  middle-aged women. Generally, the importance of this investigation was not recognized and not well deserved. The abstract does not covers strictly the main statements of publication, Introduction is too short to explain research aim, design, tool (suspension training, adlay supplementation)  and so on. And moreover, there is no research creativity, whereas this kind of research much performed. The narration of introduction dealing with the specific mechanism suspension training, adlay supplementation, arterial stiffness. It needs to be described about specific rationals and references in the section of Introduction, methods and discussions. Discussion must need to be describes in more details with many other references. 

Author Response

(The authors gave the same response as above.)

Round 2

Reviewer 2 Report

The authors addressed most of my concerns. I recommend this manuscript be accepted in its present form. 

Reviewer 3 Report

The importance of this investigation was not recognized and defined There is little research creativity and therefore there needs a rationales about the combined effects of adlay supplementation and suspension training
There needs specific rationales and references in the introduction and method section
And There needs more discussion in datails with many related references

Author Response

Response to Reviewers

Manuscript Title: Adlay consumption combined with suspension training improves blood lipids and pulse wave velocity

Editorial Board Member,

Healthcare

We appreciate the positive comments from the reviewers. Based on your comments, our manuscript was finally revised. We eagerly look forward to publishing the revised manuscript in the Healthcare.

To Reviewers:

We thank the reviewer for important suggestions on our manuscript. We carefully read each reviewer's comments and revised the manuscript accordingly. Each comment is outlined below, point by point. Therefore, we believe that the revised version of the manuscript is a significant improvement over the initial submission.

Reviewer’s comment: Generally, the importance of this investigation was not recognized and not well deserved.
Author’s Response: Thank you for your comment However, we have a hard time agreeing with you. The reason is that it is difficult for us to understand exactly where you are unworthy and that the investigation is inaccurate. A study by Samadpour Masouleh (2021, Nutrients) reported synergistic effects of suspension training combined with nutritional supplementation [, which highlights the need to combine suspension training with nutritional approaches. Based on this, we hypothesized that combining adlay intake with suspension training would have a synergistic effect on the health and fitness of middle-aged women compared to using suspension training alone. We cannot accept your opinion as baseless.

Reviewer’s comment: The abstract does not covers strictly the main statements of publication, Introduction is too short to explain research aim, design, tool (suspension training, adlay supplementation) and so on.
Author’s Response: Although it was said that the abstract does not cover the main contents of the publication, we think that it is properly composed based on the format of Introduction, Purpose, Methods, Results, and Conclusion based on MDPI guidelines. In addition, we strictly adhered to the character limit, and we cannot take your comments positively because these comments are contrary to those of other reviewers. I hope you understand this. Also, you pointed out the length of the introduction. The part recommended by publishers that publish many research papers recently is to reduce the length of the overall text while explaining the main content. Based on these latest opinions, we believe that the need for our research thesis has been adequately explained. Additionally, based on the opinions of other reviewers, we have appropriately revised the introduction section.

Reviewer’s comment: And moreover, there is no research creativity, whereas this kind of research much performed.
Author’s Response: You said that there is a lot of research of this kind, but there is no research creativity. However, although it can be said that the form of the study is general, it is not convincing that there is no creativity in the study of middle-aged women with physiological characteristics. In addition, it was conducted to achieve the purpose of research through subdivided grouping, and we are confident that we have produced research results worth introducing to the world. We do not agree with the reviewer's assertion that our study was of no value. I think these parts are very stuffy.

Reviewer’s comment: The narration of introduction dealing with the specific mechanism suspension training, adlay supplementation, arterial stiffness. It needs to be described about specific rationals and references in the section of Introduction, methods and discussions. Discussion must need to be describes in more details with many other references.

Author’s Response: We explained the basis for each sentence, and based on this, we discussed the need for research and the results.